# Synthesis, Optical Properties, and Fluorescence Cell Imaging of Novel Mixed Fluorinated Subphthalocyanines

**DOI:** 10.3390/molecules28020725

**Published:** 2023-01-11

**Authors:** Shutong Zhou, Xiaojuan Lv, Minghui Li, Zijian Gao, Shengnan Tu, Shanshan Qiao, Mengjia Mo, Xu Tang, Yemei Wang, Shasha Sun

**Affiliations:** 1School of Environmental and Chemical Engineering, Jiangsu University of Science and Technology, Zhenjiang 212013, China; 2School of Chemistry and Chemical Engineering, Jiangsu University, Zhenjiang 212013, China; 3Institute for Advanced Materials, School of Materials Science and Engineering, Jiangsu University, Zhenjiang 212013, China

**Keywords:** subphthalocyanine, peripheral substituents, theoretical calculation, fluorescence imaging

## Abstract

Subphthalocyanines (SubPcs) are a kind of tripyrrolic macrocycle with a boron atom at their core. Incorporating different units onto the SubPc periphery can endow them with various unique properties. Herein, a series of novel fluorinated low-symmetry SubPc derivatives containing chlorine groups (F_8_-Cl_4_-SubPc, F_4_-Cl_8_-SubPc) and methoxy groups (F_8_-(OCH_3_)_2_-SubPc) were synthesized and characterized by spectral methods (MS, FT-IR, ^1^H, ^13^C, ^11^B, and ^19^F NMR spectroscopy), and the effect of the peripheral substituents on their electronic structure of low-symmetry macrocycle was investigated by cyclic voltammetry, theoretical calculation, electronic absorption, and emission spectroscopy. In contrast to perfluorinated SubPcs, these low-symmetry SubPcs revealed non-degenerate LUMO and LUMO + 1 orbitals, especially F_8_-(OCH_3_)_2_-SubPc, which was consistent with the split Q-band absorptions. The cyclic voltammetry revealed that these SubPcs exhibited two or three reduction waves and one oxidation wave, which is consistent with the reported SubPcs. Finally, an intracellular fluorescence imaging study of these compounds revealed that these compounds could enter cancer cells and be entrapped in the lysosomes, which provides a possibility of future applications in lysosome fluorescence imaging and targeting.

## 1. Introduction

Subphthalocyanines (SubPcs), comprising alternately arranged three isoindole units and imino-nitrogen bridges, have been widely studied as the contracted analog of phthalocyanine (Pc) since 1972 [1]. In contrast to Pcs, the smaller central cavity of SubPcs is occupied by a tetrahedral boron atom and has an axial substituent, making them more polar and less aggregative [2,3,4]. In addition, a smaller π-electron conjugated system and cone structure make SubPcs possess excellent optical and electrochemical properties, which have been explored and applied in various fields, such as organic photovoltaics (OPVs) [5,6], photodynamic therapy (PDT) photosensitizers [7,8,9], and fluorescent ion probes [10,11,12]. Based on these structure–property relationships, the properties of SubPcs can be easily modified by peripheral substitution, axial replacement [13,14], or inner core modification [15,16]. Among these modification methods, inserting peripheral substituents is the most common and effective way to adjust the electronic structure and photoelectric properties of SubPcs. A universal way to insert various functional units into the periphery of SubPc is by utilizing phthalonitrile with different functional groups as precursors [17]. Phthalonitrile with *C*_2v_ symmetry yields *C*_3v_ symmetric SubPcs [18], whereas non-symmetric phthalonitrile starting materials yield a mixture of constitutional isomers (*C*_1_ and *C*_3_), such as 1,2-SubNc [19], binaphthyl-linked SubPc [20], and triiodo-SubPc [21]. In addition, the introduction of functional groups in the periphery of SubPc by metal-catalyzed cross-coupling reactions, Pd-catalyzed Sonogashira [22], Suzuki [23], and Stille [24] C-C bond formation reactions, or Buchwald–Hartwig amination reactions [25] have also been well explored and reported. The methods mentioned above are mainly applied to the synthesis of the SubPcs with three peripheral isoindole rings linked to the same group. Mixing multiple phthalonitrile derivatives by a usual condensation reaction in a one-pot method has always been used to produce a low-symmetry mixture SubPcs and yields a series of A_3_-, A_2_B-, AB_2_-, and B_3_-type SubPcs, where “A” and “B” are different peripheral moieties of SubPc, such as pyrene-fused SubPcs [26], fluorinated SubPcs [27], pyrazine-fused SubPcs [28], and fluorenone-fused SubPc [29]. Our group previously reported a related work that showed that fluorinated S_2_CO-fused SubPcs [30] were produced from a mixed reaction of different phthalonitrile precursors. With the change in the ratios between the peripheral substituents, the photoelectric properties of these SubPcs also showed notable changes. Although numerous low-symmetry-mixture SubPcs have been explored and reported, fluorinated SubPcs with chlorine atoms or ethoxy group have not been studied. 

Herein, we adapted previous synthetic methodologies for accessing low symmetry mixture fluorinated SubPcs containing chlorine groups (F_8_-Cl_4_-SubPc **5**, F_4_-Cl_8_-SubPc **6**) and methoxy groups (F_8_-(OCH_3_)_2_-SubPc **7**). The influence of peripheral substituents on the properties of SubPcs was investigated by combining theoretical calculation and experimentation. The systematic study of F_12_-SubPc, F_8_-Cl_4_-SubPc, and F_4_-Cl_8_-SubPc revealed that the slight difference in the composition of peripheral substituents, such as fluorine or chlorine, caused a small effect on the photoelectric properties of SubPcs. Compared with fluorinated SubPc, the absorption and emission of these mixture SubPcs showed a regular red shift with a gradual increase in peripheral chlorine atoms, and they had greater oxidation potentials. In addition, fluorinated SubPcs containing peripheral methoxy groups were also studied; reduced symmetry gave it a split Q-band absorption, and the methoxy groups enhanced its fluorescence emission. Finally, the intracellular fluorescence imaging of these three SubPcs was investigated, and the results indicated that they could be further utilized for lysosome imaging.

## 2. Results and Discussion

### 2.1. Synthesis of SubPcs

The synthetic route of the three novel SubPcs **5**–**7** is shown in Figure 1, as described in previous publications [31]. By mixing two different phthalonitrile precursors in the presence of 1 M BCl_3_ in *p*-xylene, a series of asymmetric SubPcs with different ratios of peripheral substituents were obtained. SubPcs **5** and **6** with peripheral chlorine atoms had an obviously higher solubility than SubPcs with outer methoxy groups, which is consistent with the reported SubPcs. Therefore, **5** and **6** were successfully separated and purified. The crystal structure of **6** was successfully obtained. However, the solubility of the SubPcs decreased significantly with the increased number of the outer methoxyl groups; only **7** contained two methoxyl groups and was successfully isolated and purified.

### 2.2. Mass Spectra

High-resolution atmospheric pressure chemical ionization mass spectrometry (HR-APCI-MS) confirmed the composition of the obtained low-symmetry SubPcs. The intense cluster peaks corresponding to [*M* + H]^+^ were observed at *m*/*z* = 712.86259 Da for **5** (calcd. for C_24_BCl_5_F_8_N_6_, 712.86357 Da) (Appendix A), 778.52596 Da for **6** (calcd. for C_24_BCl_9_F_4_N_6_, 778.74241 Da) (Appendix A), and 635.04375 Da for **7** (calcd. for C_26_H_8_BClF_8_N_6_O_2_, 635.04353 Da) (Appendix A). The set of peaks in the clusters was in perfect agreement with the theoretically calculated isotope distribution for corresponding compositions.

### 2.3. NMR Spectra

The ^1^H NMR spectrum was recorded in CDCl_3_ for **7** with peripheral methoxy groups; the signals of the methoxy and benzene protons were observed at 4.24 ppm and 8.28 ppm as singlets, respectively (Appendix A). In contrast to the highly symmetric **4**, which had only one single peak in the ^19^F NMR spectra, for lower symmetric **7**, four sets of resonance signals were observed as triplets at −137.66, −138.10, −148.43, and −149.27 ppm (Appendix A). Although there are free protons in **5** and **6**, the ^19^F NMR spectra were especially useful for distinguishing them. The former had two sets of multiple resonance signals for peripheral fluorine atoms at around −135.39 and −145.83 ppm (Appendix A). Differently, the later SubPc showed two sets of double resonance signals for peripheral fluorine atoms at −136.22 and −146.64 ppm, respectively (Appendix A).

The ^13^C NMR of **5**–**7** showed low-intensity signals in the low field since they belonged to the quaternary carbons with longer relaxation times (Appendix A). Comparison with the spectra of highly symmetric **4** was helpful for assigning some characteristic resonance signals. Similar to **4**, the C atoms of the CF groups in the benzene rings of **5**–**7** appeared as broad multiplets at ~145 ppm, which was caused by C-F coupling. In addition, the C_α_ and C_β_ atoms of the perfluorinated isoindole fragments in **4** appeared at around 147 ppm and 115 ppm, respectively. The C_β_ atoms of **5**–**7** appeared in similar positions, 115.18, 119.17, and 114.96 ppm, respectively, and **5**–**7** contained two inequivalent C_α_ atoms in the isoindole rings, for which two signals were observed (146.81 and 147.49 ppm for **5**; 147.69 and 148.21 ppm for **6**; 152.77 and 153.49 ppm for **7**). For **7**, a strong signal peak in the high field from the peripheral methoxy groups (56.99 ppm) was observed, while two singlet signals were derived at 152.77 and 153.49 ppm from the C atoms of the unsubstituted groups and replaced by methoxy groups in the peripheral benzene ring. The ^11^B NMR spectra of **5**–**7** contained singlets at −15.05, −14.96, and −15.08 ppm (Appendix A), which is typical for boron (III) SubPc.

### 2.4. UV/Vis Absorption and Emission

Low-symmetry **5** and **6** are readily soluble in dichloromethane, forming red-pink solutions typical of SubPcs, similar to their symmetrical analog, **4**. With the increase in chlorine atoms, the solutions of compounds **4**–**6** gradually changed from dark purple to lavender. SubPc **7** had a similar dissolution to **4** and **5** in dichloromethane, forming an orange-red solution. Their UV–Vis spectra are displayed in Figure 1a.

The absorption and emission spectra of the prepared SubPcs in dichloromethane were measured to investigate the optical properties of **4**–**7**. Like most SubPcs, **5**–**7** exhibited two major transitions, the Soret band (250–300 nm) and the Q band (550–585 nm). For the low-symmetry **5**–**7**, the maximum of the Soret band observed in the UV region at 294–303 nm was slightly shifted bathochromically compared with the symmetrical **4** (307 nm). TDDFT calculation (see Appendix A) indicated that this intense absorption in the UV region, similar to that of low-symmetry F_8_Ph_2_N_2_sPc and F_4_Ph_4_N_4_sPc [28], arose from the overlap of several complex excited states. Compared with **4**, the substitution of the peripheral fluorine atoms with the different groups had a small effect on the Q-band position and more of an effect on the Q-band shape and the fluorescence quantum yield. For **7**, the presence of the methoxy group caused the split of the Q band (Q_x_ = 559 nm, Q_y_ = 579 nm) due to its low symmetric electron structure and non-degenerate LUMO and LUMO+1 orbitals. However, **5** and **6** had the same low symmetry structure as **7**; both showed only a strong Q-absorption band, which was due to the similar electronic properties of the fluorine and chlorine atoms and the tiny ΔLUMO (the gap between LUMO and LUMO + 1). With the increase in the number of chlorine atoms, the Q band gradually shifted to the red region from **4** (573 nm) to **5** (580 nm) and **6** (585 nm). 

The fluorescence emission spectra of all studied SubPcs are shown in Figure 1b. The shifts of the emission are parallel to those observed for the Q-band absorption of **5**–**7**. The value of the Stokes shifts of 11–20 nm indicated that only small geometrical rearrangements occurred in the excited state. Probably due to the heavy atom effect [32,33], the fluorescence became quenched in the order of **5** (*Φ*_F_ = 0.014) and **6** (*Φ*_F_ = 0.010). However, the presence of the methoxy group enhanced the fluorescence of **7** (*Φ*_F_ = 0.11).

### 2.5. Crystal Structure

A crystal of **6** was obtained by slow vapor diffusion (*n*-hexane into CH_2_Cl_2_), and its structure was determined by X-ray diffraction analysis. It showed an essentially bowl-shaped structure similar to that of the general SubPcs (0.56–0.66 Å) [34], with a deviation of a boron atom by 0.601 Å from the mean plane defined by three boron-coordinating nitrogen atoms (Figure 2). The bowl depth from the boron atom to the second plane defined by peripheral halogen atoms was 2.310 Å, similar to that for the reported SubPcs (2.45–2.75 Å) [35,36]. Moreover, the average distances were B-N_p_ 1.481 Å and B-Cl 1.854 Å, which are typical for SubPc (for Cl_6_Pyz_3_sPA B–N_p_ 1.48 Å and B-Cl 1.86 Å) [37]. The length of the peripheral C-Cl bonds and C-F bonds are summarized in Appendix A. The average length of the peripheral C-Cl bond was 1.716 Å, while the peripheral C-F bond had a shorter average length of 1.345 Å, which was consistent with Cl_12_-SubPc (C-Cl 1.715 Å) [38] and F_12_-SubPc (C-F 1.343 Å) [39], respectively. In addition, the difference in the C-Cl and C-F bonds was consistent with the electronegativity difference between fluorine and chlorine atoms.

### 2.6. Electrochemistry

The redox properties of these novel SubPcs (**5**–**7**) were examined in CH_2_Cl_2_ containing 0.1 M TBAP as a supporting electrolyte at room temperature. The cyclic voltammograms of these compounds are shown in Figure 3, and the half-wave potentials and peak potentials are summarized in Table 1. One irreversible oxidation range from 1.40 to 1.52 V vs. SCE was observed for **5**–**7**, while two or three reductions were observed for them. The difference in the half-wave potentials between the first and the second reductions is given in Table 1 as ∆*E*_R_ and ranged from 0.21 to 0.59 V vs. SCE. Both **5** and **6** had one reversible reduction wave at 0.53 V vs. SCE and two irreversible reduction waves, while **7** had one reversible reduction wave and one irreversible reduction wave at −0.73 and −1.21 V, respectively.

The absolute differences between the first reduction and first oxidation (the HOMO-LUMO gap) of them ranged from 2.02 to 2.13 V (V vs. SCE), with an average value of 2.075 ± 0.055 V, which is almost the same as the value of the reported mixture SubPcs. 

### 2.7. Molecular Orbital Calculations

To study the relationship between the optical results and the electronic structures, molecular orbital (MO) calculations on the structures of **4**–**7** were carried out at the B3LYP/6-31G(d) level. The frontier MOs of **4** represented typical MOs of a symmetric SubPc, featuring degenerate LUMO and non-degenerate HOMO and HOMO − 1. In agreement with previous studies on low symmetric SubPcs, decreasing the symmetry of SubPc lifted the degeneracy of the LUMO so that **5**–**7** all exhibited non-degenerate LUMO and LUMO + 1. The time-dependent (TD) DFT calculation of **7** reproduced the observed split Q bands with transitions from the HOMO to the non-degenerate LUMO and LUMO + 1, and the calculated absorption spectra of **5** and **6** both revealed that their Q band regions were composed of the HOMO, LUMO, and LUMO + 1; LUMO and LUMO + 1 were slightly non-degenerated (Figure 4). With an increased number of peripheral chlorine atoms, the LUMO and LUMO + 1 of **5** and **6** became more stable, as well as their HOMO. However, the peripheral methoxy groups destabilized the LUMO, LUMO + 1, and HOMO of **7**. The energy gaps between the HOMO-LUMO and HOMO-LUMO + 1 of these produced SubPcs were also explored. The energy gaps of **5**–**7** were almost the same, all smaller than **4**, which was consistent with their absorption spectra.

### 2.8. Cell Fluorescence Imaging

To explore the potential application of SubPcs in cancer cell fluorescence imaging, **5**–**7** were incubated with HeLa cells, and their fluorescence distributions were monitored by fluorescence microscopy. For the presence of peripheral electron-drawing groups, **5**–**7** had high solubility. In addition, solutions of the SubPcs **5**–**7** (2.0 × 10^−6^ mol/L) were prepared in dimethyl sulfoxide (DMSO); the final concentration of DMSO was always lower than 0.5% (*v*/*v*). HeLa cells were placed in confocal culture dishes and cultured overnight in DMEM (Dulbecco’s modified Eagle’s medium, Thermo Fisher Scientific, Waltham, MA, USA) high-glucose media with 10% (*v*/*v*) fetal bovine serum (FBS). The cells were then washed three times with PBS solution. Subsequently, 10 μL quantities of SubPcs **5**–**7** were separately added to the cell culture media and cultured with HeLa cells at 37 °C. After incubation for 4 h, the cells were washed three times with PBS; the lysosomes and nuclei were then stained with LysoTracker Green DNA-26 (1 μg/mL) (Thermo Fisher Scientific) and Hoechst 33342 (1 μg/mL) for 30 min (Thermo Fisher Scientific). The staining was removed by washing with PBS buffer, and the solutions were then magnified using the 100× objective of an inverted fluorescence microscope and captured using the corresponding fluorescence channel [8,40,41].

Hoechst 33342 and LysoTracker Green DND-26 were used to label the nuclei and lysosomes of HeLa cells by emitting blue and green fluorescence, respectively. After cell incubation for a short time, the fluorescence images showed that the SubPcs were distributed in the cytoplasm of the HeLa cells with remarkable orange fluorescence (Figure 5). When merging the fluorescence images of HeLa cells in different panels, the fluorescence of the SubPcs was found to be co-localized with the green fluorescence of the lysosome marker, indicating that most of the SubPcs were trapped in lysosomes through cellular endocytosis. After incubation for 24 h, the cells remained intact with normal morphology, indicating that **5**–**7** have good biocompatibility and could be used for cancer cell labeling in the future.

## 3. Materials and Methods

F_8_-Cl_4_-SubPc **5**, F_4_-Cl_8_-SubPc **6**, and F_8_-(OCH_3_)_2_-SubPc **7** were synthesized using a common method in the literature and characterized by HR-APCI-MS, NMR, IR, and UV–Vis.

Preparation of F_8_-Cl_4_-SubPc (**5**): A mixture of phthalonitrile precursors of **1** (0.40 g, 2.0 mmol) and **2** (0.27 g, 1.0 mmol) was added to a Schlenk tube, and the tube was filled with nitrogen gas. A 1.0 M *p*-xylene solution of BCl_3_ (3.0 mL, 3.0 mmol) was added, and the reaction mixture was stirred under reflux for 3 h. The end of the reaction was controlled by the appearance of several purple spots on the TLC, the first of which was **5** (R*_f_* = 0.43, in CH_2_Cl_2_/Hexane = 1/2). In addition, a dark violet mixed solvent was obtained at the end of the reaction. The solvent was removed by evaporation under reduced pressure, and the product was directly purified by silica gel column chromatography; recrystallization from CH_2_Cl_2_ and C_6_H_14_ provided **5** as a dark purple solid with a golden glow at a 3.2% yield (23 mg, 0.032 mmol). HRMS (APCI) (*m/z*): 712.86259 Da (calcd. for C_24_BCl_5_F_8_N_6_ = 712.86357 Da [*M*]^+^); ^19^F NMR (CDCl_3_, 376 MHz, 298 K): *δ* = −135.39 (d, *J_o_* = 13.59 Hz, 4F), −145.83 (s, 4F) ppm; ^13^C NMR (CDCl_3_, 100 MHz, 298 K): *δ* = 115.18, 127.54, 128.98, 141.45, 143.88, 144.36, 146.81, 147.49 ppm; ^11^B NMR (CDCl_3_, 128 MHz, 298 K): *δ* = −15.05 ppm; FT-IR (CH_2_Cl_2_), *ν* (cm^−1^): 2956, 2921, 2851, 2359, 1714, 1531, 1492, 1458, 1409, 1259, 1117, 1081, 1015, 972, 859, 709, 538, 498, 404; UV-Vis (CH_2_Cl_2_): *λ*_max_ (*ε*) = 303 (14,575), 580 (13,750) nm; M. p. > 250 °C.

Preparation of F_4_-Cl_8_-SubPc (**6**): A mixture of phthalonitrile precursors of **1** (0.40 g, 2.0 mmol) and **2** (0.27 g, 1.0 mmol) was added to a Schlenk tube, and the tube was filled with nitrogen gas. A 1.0 M *p*-xylene solution of BCl_3_ (3.0 mL, 3.0 mmol) was added, and the reaction mixture was stirred under reflux for 3 h. The end of the reaction was controlled by the appearance of several purple spots on the TLC, the second of which was **6** (R*_f_* = 0.29, in CH_2_Cl_2_/Hexane = 1/2). In addition, a dark violet mixed solvent was obtained at the end of the reaction. The solvent was removed by evaporation under reduced pressure, and the product was directly purified by silica gel column chromatography; recrystallization from CH_2_Cl_2_ and C_6_H_14_ provided **6** as a dark purple solid with a golden glow at a 7.2% yield (28 mg, 0.036 mmol). HRMS (APCI) (*m/z*): 778.52596 Da (calcd. for C_24_BCl_9_F_4_N_6_ = 778.74241 Da [*M*]^+^); ^19^F NMR (CDCl_3_, 376 MHz, 298 K): *δ* = −136.22 (d, *J_o_* = 15.04 Hz, 2F), −146.64 (d, *J_o_* = 15.04 Hz, 2F) ppm; ^13^C NMR (CDCl_3_, 100 MHz, 298 K): *δ* = 119.17, 124.09, 124.56, 129.58, 134.31, 136.22, 138.52, 147.21, 147.69 ppm; ^11^B NMR (CDCl_3_, 128 MHz, 298 K): *δ* = −14.96 ppm; FT-IR (CH_2_Cl_2_), *ν* (cm): 2956, 2917, 2849, 2359, 1739, 1493, 1462, 1399, 1362, 1187, 1080, 1050, 967, 404; UV-Vis (CH_2_Cl_2_): *λ*_max_ (*ε*) = 296 (10,150), 585 (17,175) nm; M. p. > 250 °C.

Preparation of F_8_-(OCH_3_)_2_-SubPc (**7**): A mixture of phthalonitrile precursors of **1** (0.25 g, 1.25 mmol) and **3** (0.12 g, 0.63 mmol) was added to a Schlenk tube, and the tube was filled with nitrogen gas. A 1.0 M *p*-xylene solution of BCl_3_ (1.9 mL, 1.9 mmol) was added, and the reaction mixture was stirred under reflux for 2 h. The end of the reaction was controlled by the appearance of several magenta spots on the TLC, the second of which was **7** (R*_f_* = 0.57, in CH_2_Cl_2_/Hexane = 1/1). In addition, a dark purplish red mixed solvent was obtained at the end of the reaction. The solvent was removed by evaporation under reduced pressure, and the product was directly purified by silica gel column chromatography; recrystallization from CH_2_Cl_2_ and C_6_H_14_ provided **7** as an amaranth solid with a golden glow at an 8.6% yield (34 mg, 0.054 mmol). HRMS (APCI) (*m/z*): 635.04375 Da (calcd. for C_26_H_8_BClF_8_N_6_O_2_ = 635.04353 Da [*M*]^+^); ^1^H NMR (CDCl_3_, 400 MHz, 298 K): *δ* = 8.28 (s, 2H, methoxyl), 4.24 (s, 6H, a-benzo) ppm; ^19^F NMR (CDCl_3_, 376 MHz, 298 K): *δ* = −137.66 (t, *J_o_* = 37.6 Hz, 2F), −138.10 (t, *J_o_* = 37.6 Hz, 2F), −148.43 (t, *J_o_* = 37.6 Hz, 2F), −149,27 (t, *J_o_* = 37.6 Hz, 2F) ppm; ^13^C NMR (CDCl_3_, 100 MHz, 298 K): *δ* = 56.99 (-OCH_3_), 103.83, 114.97, 125.67, 140.63, 141.32, 143.13, 143.97, 144.58, 146.56, 152.77, 153.49 ppm; ^11^B NMR (CDCl_3_, 128 MHz, 298 K): *δ* = −15.08 ppm; FT-IR (CH_2_Cl_2_), *ν* (cm^−1^): 2988, 2901, 2360, 2342, 1051, 669, 449, 425, 411; UV-Vis (CH_2_Cl_2_): *λ*_max_ (*ε*) = 294 (11,666), 559 (12,741), 579 (15,744) nm; M. p. > 250 °C.

## 4. Conclusions

In summary, novel SubPcs **5**–**7** were synthesized, and their molecular structures were characterized by HR-APCI-MS, NMR, FT-IR, and X-ray crystallography. The obtained SubPcs showed less symmetric and bowl-shaped structures. Although varied fluorinated or chlorinated SubPcs have been reported, SubPcs **5** and **6**, which contain both chlorine and fluorine atoms at their periphery, were reported here for the first time. Their optical properties indicate that the slight distinction between the peripheral groups caused small changes in the optical properties of the SubPcs. As the peripheral fluorine atoms were gradually replaced by chlorine atoms, the absorption and fluorescence of SubPcs **4**–**6** gradually redshifted by less than 13 nm. However, the properties of the peripheral substituents had a great influence on the fluorescence quantum yield of the SubPcs. In addition, the C-F length of SubPc **6** was similar to that of the C-F in the perfluorinated SubPc **4**, and its C-Cl length was consistent with that of the C-Cl in the perchlorinated SubPc, indicating that the reduction in symmetry had little effect on the distance between carbon atoms and halogens in the peripheral benzene ring.

The cyclic voltammograms of **5**–**7** indicated two or three reduction waves and one oxidation wave. Compared with the oxidation of perfluorinated **4**, the oxidation process of **5** and **6** gradually shifted to positive with the increase in the number of peripheral chlorine atoms. In addition, **5**–**7** had very small differences in their redox gaps (~2.05 V), which was in accordance with their similar Q-band absorptions and HOMO-LUMO gaps. According to the analysis of the frontier molecular orbitals by theoretical calculation, the peripheral chlorine atoms with less electronegativity caused the non-degenerate LUMO orbitals and LUMO + 1 orbitals and slightly stabilized the HOMO orbitals. However, the introduction of the methoxy group led to the non-degenerate LUMO and LUMO + 1 orbitals of **7**, which also destabilized the LUMO and HOMO orbitals. In addition, the subcellular localization of **5**–**7** was further examined by fluorescence microscopy; the results showed that **5**–**7** had high solubility and could be internalized readily into cancer cells and localized in lysosomes in a short time, which implies that they have great potential in biological cancer cell imaging.

## Data Availability

Not applicable.

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
