# Peer review of "Synthesis, Optical Properties, and Fluorescence Cell Imaging of Novel Mixed Fluorinated Subphthalocyanines"

_molecules, 2023, doi:10.3390/molecules28020725_

Round 1
Reviewer 1 Report
In this manuscript titled by "Synthesis, Optical Properties and Fluorescence Cell Imaging of Novel Mixed Fluorinated Subphthalocyanines " - the methods of MS, 1H and 19F NMR spectroscopy for structure and spectral characteristic ovel fluorinated low-symmetry SubPc derivatives containing chlorine groups and methoxy groups is used. The contents of the reviewed manuscript are very well described by the title. The authors show that these compounds could enter cancer cells and entrap in the lysosome.
In my point, this manuscript is well-written and could be recommended for publication in Molecules after a coresponding revision in which the following issues are addressed:
-The innovative of this study was not enough. Please add more discription and comperison for similar compounds to understand the novelty of the study.
[10.1016/j.jphotochem.2022.114404, 10.1016/j.dyepig.2022.110282, 10.1016/j.dyepig.2020.108944, 10.1016/j.dyepig.2019.03.010].
- In the part of the NMR experiment, information on the chemical shift standard for 1H and 19F should be added.
- Additionally, for NMR experiments, the parameters of the NMR spectrometer equipment and the pulse sequence used must be discription.
-It is necessary to obtain 11B NMR spectra to confirm the chemical structure of low-symmetry SubPc.
- In suplementary materials you must indicate the PPM in x-axes for all NMR spectra.
- The study of the structure by X-ray must be confirmed by comparison with the crystallographic database for similar compound or experimintal spectra evidence.
-The authors have to shed light on the similarities and differences among their work and the literatures of the problem. A clear explanation, what is the new result in their work, and how it is build up upon previous work in the field. For example to many componds can show intracellular fluorescence how is your compounds better?
-A conclusion section should be extended to include more details
If the authors submit a new version according to my suggestions where they also give more explanations about the experimental and ather problems, I could recommend the paper for publication.
Author Response
Point 1: The innovative of this study was not enough. Please add more discription and comperison for similar compounds to understand the novelty of the study. [10.1016/j.jphotochem.2022.114404, 10.1016/j.dyepig.2022.110282, 10.1016/j.dyepig.2020.108944, 10.1016/j.dyepig.2019.03.010].
Response 1: Thank you for your comment and recommended articles. We have carefully read the articles you recommended and others, and compared our samples with similar SubPcs to give a more detailed description of the UV, NMR, crystal structure and cell experiment of them.
In the section of UV absorption, we described the color of the solution of SubPcs 5-7. Compared with the low symmetry SubPcs F8Ph2N2sPc and F4Ph4N4sPc, we revealed the intense absorption in the UV region arises from overlap of several complex excited states based on the theory calcilation. (Lines 120-131, Page 3)
In the section of NMR, we have added the 11B and 13C NMR spectra of these resulting SubPcs to further know our samples. (Figure S6, S7, S9, S10, S12 and S13 in SI)
In the section of crystal structure, the crystal structure of SubPc 6 was further analyzed by comparing with that of F12SubPc, Cl12SubPc and Cl6Pyz3sP. The length of B−Np and axial B-Cl of SubPc 6 was consistent with that of conventional SubPcs. The C−F length is similar with that of the C−F in perfluorinated SubPc 4, and its C−Cl length is consistent with that of the C−Cl in perchlorinated SubPc, indicating that the reduction of symmetry has little effect on the distance between carbon atoms and halogens in the peripheral benzene ring. (Lines 156-163, Page 4)
In the section of cell experiment, we have added an adequate level of detail, and appropriate references on the methods and materials used. (Lines 211-224, Page 7)
Point 2: In the part of the NMR experiment, information on the chemical shift standard for 1H and 19F should be added.
Response 2: Thank you for your comment. The information on the chemical shift standard for 1H and 19F have been added in the general information of SI.
Point 3: Additionally, for NMR experiments, the parameters of the NMR spectrometer equipment and the pulse sequence used must be discription.
Response 3: Thank you for your comment. The parameters of the NMR spectrometer equipment and the pulse sequence used have been discribed in the general information of SI.
Point 4: It is necessary to obtain 11B NMR spectra to confirm the chemical structure of low-symmetry SubPc.
Response 4: Thank you for your comment about the 11B NMR spectra. 11B NMR spectra of 5-7 in CDCl3 have been supplemented in supporting information (Figure S7, S10 and S13). The signals of 11B NMR spectra are described in detail in the NMR spectra section of the revised manuscript (Lines 117-118, Page 3).
Point 5: In suplementary materials you must indicate the PPM in x-axes for all NMR spectra.
Response 5: Thank you for your comment. All NMR spectra have been corrected in SI.
Point 6: The study of the structure by X-ray must be confirmed by comparison with the crystallographic database for similar compound or experimintal spectra evidence.
Response 6: Thank you for your comment. As the response 1, the crystal structure of SubPc 6 was further analyzed by comparing with that of F12SubPc, Cl12SubPc and Cl6Pyz3sP. The length of B−Np and axial B-Cl of SubPc 6 was consistent with that of conventional SubPcs. The C−F length is similar with that of the C−F in perfluorinated SubPc 4, and its C−Cl length is consistent with that of the C−Cl in perchlorinated SubPc, indicating that the reduction of symmetry has little effect on the distance between carbon atoms and halogens in the peripheral benzene ring. (Lines 156-163, Page 4)
Point 7: The authors have to shed light on the similarities and differences among their work and the literatures of the problem. A clear explanation, what is the new result in their work, and how it is build up upon previous work in the field. For example to many componds can show intracellular fluorescence how is your compounds better?
Response 7: Thank you for your comments. The main innovation of this paper is that we have synthesized and characterized a series of novel mixed SubPcs. We noted in the paper that although varied fluorinated or chlorinated SubPcs have been reported. SubPcs 5 and 6, which contain both chlorine and fluorine atoms on their periphery, have been reported here for the first time. By comparing our mixtured SubPcs with perfluorinated and perchlorinated SubPcs, the effect of symmetry reduction on properties was obtained. In addition, our cell experiments revealed that our compound was able to enter cells within a short incubation period (4 h) and maintain cell normal morphology within 24 hours, which provided the possibility for further cell experiments such as photodynamic therapy and photothermal therapy in our later period.
Point 8: A conclusion section should be extended to include more details
Response 8: Thank you for your comments. We've added more details on crystal structure, electrochemistry and cell experiment of compounds in the conclusion section. (Lines 295-322, Page 8-9).
Reviewer 2 Report
The study on modified SubPcs is attractive due to the interesting optical, structure and electronic behaviors as well as the application in many fields. In this manuscript, the authors prepared three novel SubPcs 5-7 which were characterized in detail by 1H, and 19F NMR spectra and HRMS. Importantly, the structure of 6 was revealed by X-ray diffraction analysis. The photoelectric properties of these mixed SubPcs were investigated by UV-vis absorption/fluorescence spectrum, cyclic voltammograms spectrum, and TD-DFT calculation. Finally, the authors found that 5-7 have good biocompatibility and could be used for cancer cell labeling in the future.
This work will attract readers who work in the fields of organic functional material science, dyes and pigments science, and supramolecular chemistry. I recommend it to be published in Molecules as an article after minor revision.
(1) Providing the11B NMR spectra of these resulting SubPcs is necessary.
(2) In the part of mass spectra,"calc" were used for SubPc 5 and 6, while "calcd" for SubPc 7. Please use “calcd”.
(3) Line 124, please change “its structure was determined by X-ray diffraction” to “its structure was determined by X-ray diffraction analysis”
(4) Please recheck the 19F spectra and give the coupling constant (J).
(5) Please use the same expression for the name journal, such as “Angew. Chem. Int. Ed.” in refs.15, 16 and 30, not “Angew. Chem. Int. Ed. Engl”.
(6) In the part of “Materials and Methods”, if the number represents the corresponding compound, please use boldface.
(7) Are you sure the size of silica gel you used is 300-400 μm, not 300-400 mesh?
Author Response
Point 1: Providing the 11B NMR spectra of these resulting SubPcs is necessary.
Response 1: Thank you for your comment about the 11B NMR spectra. 11B NMR spectra of 5-7 in CDCl3 have been supplemented in supporting information (Figure S7, S10 and S13). The signals of 11B NMR spectra are described in detail in the NMR spectra section of the revised manuscript (Lines 117-118, Page 3)
Point 2: In the part of mass spectra,"calc" were used for SubPc 5 and 6, while "calcd" for SubPc 7. Please use “calcd”.
Response 2: Thank you for your comment about the spelling of English words of the manuscript. All the corrections mentioned above have been implemented revised. (Lines 90-91 Page 3, 254, 270, and 285 Page 8)
Point 3: Line 124, please change “its structure was determined by X-ray diffraction” to “its structure was determined by X-ray diffraction analysis”
Response 3: Thank you for your comment. We have replaced “its structure was determined by X-ray diffraction” with “its structure was determined by X-ray diffraction analysis”.(Lines 153, Page 4)
Point 4: Please recheck the 19F spectra and give the coupling constant (J).
Response 4: Thank you for your comment about the 19F NMR spectra.I have recheck the 19F spectra and give the coupling constant (J) of SubPcs 5-7, respectively. (Lines 255, 271 and 287, Page 8)
Point 5: Please use the same expression for the name journal, such as “Angew. Chem. Int. Ed.” in refs.15, 16 and 30, not “Angew. Chem. Int. Ed. Engl”.
Response 5: Thank you for your comments on the References section. We replace all “Angew. Chem. Int. Ed. Engl” in reference with “Angew. Chem. Int. Ed.” in ref. 13, 15, 16, 30 and 39.
Point 6: In the part of “Materials and Methods”, if the number represents the corresponding compound, please use boldface.
Response 6: Thank you for reminding. We've bolded all the numbers that represent compounds in the section of Materials and Methods. (Page 7-8)
Point 7: Are you sure the size of silica gel you used is 300-400 μm, not 300-400 mesh?
Response 7: Thank you for pointing out our mistake. We have replaced “300-400 μm” with “300-400 mesh” in SI.
Reviewer 3 Report
Dear Authors,
The research submitted is interesting, but I would like to do some remarks on issues which need to be solved to improve the manuscript first.
Please bear in mind that the experimental methods have to be as detailed as to allow any chemist to repeat the described procedures. More experimental detail on the preparation and purification of the products must be provided.
A new compound must be characterized in full, and that includes proof of its identity, melting points, elemental analysis, spectral data, including relevant IR bands, 1H, 19F and 13C NMR.
In lines 216 and 225 it is stated that compounds 6 and 7 are "dark light purple". Please correct.
Compounds 5, 6 and 7 are solid, so you can obtain their melting points.
UV spectra must state which solvent was used, and also concentration.
The cell imaging part needs an adequate level of detail, and appropriate references on the methods and materials used.
In line with these comments, I think the manuscript needs a Major revision before proceeding any further.
Author Response
Point 1: Please bear in mind that the experimental methods have to be as detailed as to allow any chemist to repeat the described procedures. More experimental detail on the preparation and purification of the products must be provided.
Response 1: Thank you very much for your guidance. We have added more experimental details on the preparation and purification of the products in Materials and Methods section.(Page 7-8)
Point 2: A new compound must be characterized in full, and that includes proof of its identity, melting points, elemental analysis, spectral data, including relevant IR bands, 1H, 19F and 13C NMR.
Response 2: Thank you for your comment. Melting points, IR , 11B and 13C NMR were supplemented in the manuscript and SI. We have also done elemental analysis of the compounds, but the Cl atoms and the F atoms are often not detected by our instruments. The presence of F and Cl atoms in the compounds have been determined by the 19F NMR and crystal structure, so we do not think that elemental analysis is necessary here.
Point 3: In lines 216 and 225 it is stated that compounds 6 and 7 are "dark light purple". Please correct.
Response 3: Thank you for your kind remind. I'm sorry that I didn't describe it properly here. What I want to express is purple solid with a glossy sense. We took a closer look at three SubPcs and more accurately described the colors of their solutions and solids in revised manuscript. (Lines 253, 269 and 284, Page 8)
Point 4: Compounds 5, 6 and 7 are solid, so you can obtain their melting points.
Response 4: Thank you for your comment. Melting point of samples were recorded on a melting point tester of Tianfen-RY-51. The melting point tester we used had a large error above 250 ℃, so we stopped the test when the temperature reached 250 ℃. The SubPcs 5-7 does not begin to melt at 250 ℃, so the melting point of our compounds is recorded as M. P. > 250 ℃, which is consistent with the reported SubPcs. They were added in materials and methods section. (Lines 260, 275 and 292, Page 8)
Point 5: UV spectra must state which solvent was used, and also concentration.
Response 5: Thank you for your comments. UV spectra has been corrected (Figure 1).
Point 6: The cell imaging part needs an adequate level of detail, and appropriate references on the methods and materials used.
Response 6: Thank you for reminding. We've added more details on cell cultures, photosensitizer incubation and subcellular localization:of cell experiment. And three ppropriate references on the methods and materials used were supplemented.(Lines 211-224, Page 7)
Round 2
Reviewer 3 Report
Dear Authors,
I am glad to inform that I recommend the manuscript for publication in its present form.
Best regards